# Activation of ADRB2/PKA Signaling Pathway Facilitates Lipid Synthesis in Meibocytes, and Beta-Blocker Glaucoma Drug Impedes PKA-Induced Lipid Synthesis by Inhibiting ADRB2

**DOI:** 10.3390/ijms23169478

**Published:** 2022-08-22

**Authors:** Ikhyun Jun, Young Joon Choi, Bo-Rahm Kim, Kyoung Yul Seo, Tae-im Kim

**Affiliations:** 1The Institute of Vision Research, Department of Ophthalmology, Yonsei University College of Medicine, 50-1 Yonsei-ro, Seodaemungu, Seoul 03722, Korea; 2Corneal Dystrophy Research Institute, Yonsei University College of Medicine, 50-1 Yonsei-ro, Seodaemungu, Seoul 03722, Korea; 3Department of Ophthalmology, Ajou University School of Medicine, 164, World cup-ro, Yeongtong-gu, Suwon 16499, Korea

**Keywords:** protein kinase A, ADRB2, lipogenesis, PPARγ, meibomian gland dysfunction, timolol

## Abstract

Meibomian gland dysfunction is one of the main causes of dry eye disease and has limited therapeutic options. In this study, we investigated the biological function of the beta 2-adrenergic receptor (ADRB2)/protein kinase A (PKA) pathway in lipid synthesis and its underlying mechanisms in human meibomian gland epithelial cells (HMGECs). HMGECs were cultured in differentiation media with or without forskolin (an activator of adenylate cyclase), salbutamol (an ADRB2 agonist), or timolol (an ADRB2 antagonist) for up to 4 days. The phosphorylation of the cAMP-response element-binding protein (CREB) and the expression of peroxisome proliferator activator receptor (PPAR)γ and sterol regulatory element-binding protein (SREBP)-1 were measured by immunoblotting and quantitative PCR. Lipid synthesis was examined by LipidTOX immunostaining, AdipoRed assay, and Oil Red O staining. PKA pathway activation enhanced PPARγ expression and lipid synthesis in differentiated HMGECs. When treated with agonists of ADBR2 (upstream of the PKA signaling system), PPARγ expression and lipid synthesis were enhanced in HMGECs. The ADRB2 antagonist timolol showed the opposite effect. The activation of the ADRB2/PKA signaling pathway enhances lipid synthesis in HMGECs. These results provide a potential mechanism and therapeutic target for meibomian gland dysfunction, particularly in cases induced by beta-blocker glaucoma drugs.

## 1. Introduction

Meibomian gland dysfunction (MGD) is a major cause of evaporative dry eye disease and is characterized by terminal duct obstruction and/or qualitative/quantitative changes in glandular secretion [1]. The meibomian gland secretes lipid-rich substances called meibum to the ocular surface, which stabilizes the tear film by preventing its evaporation. Conversely, MGD destabilizes the homeostasis of the tear film and elicits dry eye-related ocular symptoms, followed by the deterioration of the patient’s visual function and quality of life [2,3].

Acinar cells of meibomian glands (meibocytes) mainly produce meibum and maturate from proliferating, non-lipid-producing basal cells to non-proliferating, lipid-accumulating differentiated cells. Lipid biosynthesis and accumulation are hallmarks of meibocyte maturation. It has been reported that various signal pathways are involved in this maturation process. Notch signaling enhances human meibomian gland epithelial cell (HMGECs) differentiation and lipid production [4]. The loss of the CD147-mediated pathway was associated with a lower number of acini and a decrease in lipid-filled meibocytes in CD147 knockout mice [5]. The hedgehog signaling pathway has an antagonistic effect on the proliferation and differentiation of rat meibomian gland epithelial cells [6]. Transcriptome analysis revealed multiple pathways, such as the insulin signaling pathway, mammalian target of rapamycin pathway, and peroxisome proliferator activator receptor (PPAR) signaling, which are possibly associated with lipid synthesis in HMGECs and mouse meibocytes [7,8]. PPARγ is a member of the nuclear receptor superfamily and is considered a master regulator of adipogenesis [9,10]. In the case of the meibomian gland, the PPARγ agonist was shown to enhance maturation and lipid synthesis in HMGECs and mouse meibocytes, and our group previously reported that interkeukin-4 facilitated lipogenesis in HMGECs via the STAT6/PPARγ signaling pathway [11,12,13]. However, there are no studies on the role of the protein kinase A (PKA) pathway in the differentiation and lipogenesis of meibocytes.

PKA, a tetrameric enzyme consisting of two catalytic subunits, each bound to a regulatory subunit dimer [14], is known to be the primary target for cyclic adenosine monophosphate (cAMP) in the cell and plays a role as the principal effector mechanism for G-protein-coupled receptors linked to adenylate cyclase [15]. In adipocytes, it is well-known that PKA phosphorylation and the activation of downstream molecules, such as the cAMP-response element-binding protein (CREB), are essential for the transcription of PPARγ and subsequent adipogenesis [16]. In ophthalmology, the PKA pathway has been mainly studied for its role in aqueous humor flow and neuroprotection in glaucoma [17]. Regarding dry eye syndrome, although one study reported that the cAMP/PKA pathway activated by α-melanocyte-stimulating hormone has a protective effect in a dry eye rat models [18], no study has investigated the role of the PKA signaling pathway in the meibomian gland.

In addition, it is clinically important to study the PKA pathway because eye drops that modulate the PKA pathway are widely used in glaucoma patients. Beta blockers are one of the most widely used anti-glaucoma eye drops and are known to cause damage to the ocular surface, including MGD [19,20,21]. However, the pathophysiology of MGD-inducing effect is remains to be elucidated.

We hypothesized that the PKA signaling pathway plays a role in the differentiation and lipid accumulation of meibocytes, similar to adipocytes, and we examined the role of the PKA pathway in the differentiation and lipid accumulation of meibocytes using HMGECs. We further investigated the function of adrenergic receptors, which could affect the PKA pathway and meibocyte maturation.

## 2. Results

### 2.1. PKA Activation Induced Lipogenesis in Human Meibomian Gland Epithelial Cells (HMGECs)

CREB phosphorylation was assessed to evaluate the PKA pathway in HMGECs. HMGECs were treated with forskolin, which activates adenylyl cyclase, thereby increasing the intracellular concentration of cAMP and activating the PKA pathway. The phosphorylation of CREB at the ser133 residue was assessed by immunoblotting at 0, 15, 30, 60, and 120 min. The phosphorylation of CREB in HMGECs increased after 15 min and gradually decreased after 120 min. We then treated HMGECs with escalating doses of forskolin to HMGECs and the level of phosphorylated CREB (pCREB) was examined at 30 min. As shown in Figure 1A, the pCREB levels increased in a dose-dependent manner. These results indicate that the PKA pathway is well conserved in HMGECs.

It is generally accepted that cAMP signaling, followed by PKA activation, plays an important role in lipogenesis in white adipose tissue [16]. To determine the role of the cAMP/PKA pathway in lipogenesis in HMGECs, we treated HMGECs with forskolin and examined the expression levels of PPARγ and sterol regulatory element-binding protein (SREBP-1), the master regulators of lipogenesis. Cells were treated with forskolin on differentiation days 0 and 2, and PPARγ and SREBP-1 were examined on days 0, 2, and 4. Forskolin-treated HMGECs showed an enhanced expression of PPARγ and SREBP-1 at both the mRNA and protein levels compared to control HMGECs (Figure 1B,C). Lipid synthesis was measured by AdipoRed assay and LipidTOX immunostaining in differentiated HMGECs on day 4, with or without forskolin, and lipids accumulated more in forskolin-treated HMGECs than in the untreated controls (Figure 1D). Overall, the activation of the cAMP/PKA pathway in differentiated HMGECs promoted additional lipid synthesis and lipid droplet accumulation.

### 2.2. Differentiated HMGECs Express Beta 2-Adrenergic Receptor

Diverse hormones and extracellular signals utilize cAMP-dependent signal transduction to exert their effects; sympathetic activation of beta-adrenergic receptors by catecholamines is a classic example [22]. Therefore, we assessed whether beta 2-adrenergic receptors (ADRB2) were expressed in HMGECs. The mRNA expression of ADRB2 increased gradually as HMGECs differentiated (Figure 2A). The protein expression of ADRB2 was scarcely detected in pre-differentiated HMGECs, but, like mRNA expression, progressively increased as HMGECs differentiated (Figure 2B), suggesting that lipid synthesis, governed by the cAMP/PKA pathway, could be regulated via ADRB2 signals.

### 2.3. Beta 2-Adrenergic Agonist Facilitates PPARγ Expression and Lipid Droplet Synthesis

To investigate the influence of ADRB2 signal cues on lipid synthesis and PPARγ expression levels, we first evaluated the effect of salbutamol, a selective beta 2-adrenergic receptor agonist, on the PKA signaling pathway in differentiated HMGECs. Salbutamol was added to differentiated HMGECs and the phosphorylation of CREB was assessed at a series of time points, then at 10 min each with escalating doses. Figure 3A demonstrated that Salbutamol enhanced CREB phosphorylation in HMGECs in a dose-dependent manner (Figure 1). Subsequently, cells were treated with salbutamol on days 0 and 2, and the expression of PPARγ and SREBP-1 was examined by immunoblotting and quantitative PCR (qPCR) on days 0, 2, and 4. Salbutamol augmented the protein expression of PPARγ and SREBP-1 on days 2 and 4, compared to the control (Figure 3B). The transcription of PPARγ and SREBP-1 showed similar patterns to immunoblotting, although PPARγ did not reach statistical significance (Figure 3C). Lipogenesis was measured by AdipoRed assay and LipidTOX immunostaining on day 4, and salbutamol remarkably enhanced lipid accumulation in HMGECs (Figure 3D,E).

### 2.4. Beta 2-Adrenergic Antagonist Inhibits Lipogenesis in HMGECs

The finding that the ADRB2 receptor agonist facilitated the lipogenesis of differentiated HMGECs led us to examine the effect of the ADRB2 receptor antagonist on HMGECs. Timolol, a beta-adrenergic receptor blocking agent and one of the representative anti-glaucoma therapeutics, was chosen and used to differentiate HMGECs. It is well-known that long-term use of topical antiglaucoma eye drops, including timolol, is associated with meibomian gland dysfunction [19,20]. Zhang et al. reported the influence of timolol on HMGECs, stating that timolol causes a dose-dependent decrease in the survival of HMGECs and that physiological doses of timolol (0.0002%) do not influence HMGEC survival [23].

We treated HMGECs with physiological doses of timolol [24] on differentiation days 0 and 2, and PPARγ and SREBP-1 were examined on days 0, 2, and 4. The protein level of PPARγ decreased with timolol administration on days 2 and 4, and the protein level of SREBP-1 decreased on day 4 (Figure 4A). However, the mRNA expression of PPARγ and SREBP-1 was not significantly different between the timolol-treated and control groups (Figure 4B).

The lipogenesis of differentiated HMGECs was compared at day 4, and timolol-treated HMGECs showed a significantly reduced lipid droplet accumulation, as evaluated by the AdipoRed assay and LipidTOX immunostaining (Figure 4C,D).

Finally, lipid synthesis was confirmed by Oil Red O staining at day 4 after forskolin. HMGECs with differentiation media were treated with salbutamol or timolol at day 0 and 2. Consistent with previous results, HMGECs treated with forskolin and salbutamol demonstrated increased lipid accumulation. The HMGECs treated with timolol showed a significant reduction in lipid accumulation (Figure 5).

## 3. Discussion

In this study, we demonstrated that the PKA pathway plays an important role in HMGEC differentiation and lipid accumulation. Forskolin, a PKA pathway activator, enhances both differentiation and lipid synthesis in HMGECs. We then evaluated the function of ADRB2, which is a G-protein-coupled membrane receptor and an upstream signal transducer of the PKA pathway, in the differentiation of HMGECs. Salbutamol, an ADRB2 agonist, successfully induced the activation of the PKA pathway and subsequent PPARγ expression and lipid synthesis, whereas timolol showed the opposite effect. These results suggest that the PKA pathway might be related to the pathophysiology of MGD and that drugs affecting the PKA signaling pathway could have an effect on MGD.

PKA includes two regulatory subunits and two catalytic subunits [14]. When cAMP binds to the inactive PKA tetramer for both regulatory subunits, the catalytic subunit is dissociated and activated, resulting in the phosphorylation of the serine and threonine residues of the matrix protein. PKA-anchored proteins (AKAPs) can bind to the cytoskeleton or organelles and PKA regulatory subunits, thereby modulating the distribution of PKA in cells and allowing PKA to catalyze specific targets [25]. When activated, PKA phosphorylates substrates such as CREB, Raf, Bad, and GSK3, and then regulates gene expression [25]. For CREB, a well-characterized PKA substrate, when the CREB ser133 residue is phosphorylated, CREB binds to the cAMP response element and initiates target gene transcription [26,27].

The role of the PKA pathway in meibomian gland dysfunction and dry eye disease is not well-understood. Ru et al. reported the protective effects of α-melanocyte-stimulating hormone against dry eye using a scopolamine-induced dry eye rat model, and that these protective effects were mediated by the PKA-CREB pathway [18]. However, they evaluated the cornea and conjunctiva, but not the meibomian glands. Qu et al. reported the antagonistic role of hedgehog signaling in the proliferation and differentiation of rat MGECs, suggesting that the PKA pathway might play a role in the differentiation of MGECs because the PKA pathway is an antagonist of hedgehog signaling [6]. In this study, we treated HMGECs with forskolin to increase cAMP levels and evaluated CREB phosphorylation to elucidate the role of the PKA pathway in lipid synthesis in meibocytes. As shown in Figure 1, we demonstrated that the PKA pathway is well-conserved in HMGECs and promotes PPARγ expression and lipid synthesis.

The PKA pathway is known to play an important role not only in lipid synthesis but also in lipolysis of adipocytes, in connection with cAMP signals [16]. cAMP is one of the earliest discovered second messengers that plays essential roles in cell signaling and regulates many physiological processes. cAMP can regulate the transcription of various target genes, mainly through PKA and its downstream effectors, such as CREB [28]. In adipogenesis, CREB knockdown blocks the expression of PPARγ, and ectopic expression of constitutively active forms of CREB is sufficient to initiate adipogenesis in mouse 3T3-L1 adipocytes [29]. In addition, CREB promotes the expression of PPARγ, a master regulator of adipogenesis, and enhances lipid synthesis and adipogenesis [30]. In contrast, the cAMP/PKA pathway is also related to lipolysis. PKA induces the sequential breakdown of triacylglycerides through the phosphorylation of proteins involved in lipolysis, such as perilipin, adipose triglyceride lipase (ATGL), and hormone-sensitive lipase (HSL) [31,32]. The activation of the PKA pathway can promote both lipid synthesis and lipolysis, and these two opposite actions are precisely regulated by compartmentalization [16]. Optical atrophy 1 (OPA1), an AKAP, moves PKA to a separate intracellular compartment close to perilipin, and perilipin, phosphorylated by PKA, induces lipolysis [33]. In our study, the activation of the PKA pathway enhanced PPARγ expression and lipid synthesis rather than lipolysis in HMGECs. Although additional studies are needed to provide an accurate answer, this may occur due to the differentiation characteristics of the HMGEC lines used in our experiments. The HMGEC line was established by immortalizing the HMGECs of a 58-year-old male donor with retroviral human telomerase reverse transcriptase in 2010 [34], and HMGECs have been used as experimental models to mimic human meibocytes. However, it is still unclear whether the serum-induced differentiation of HMGECs reaches a late differentiation stage [35]. The possibility that the activation of the PKA pathway induces lipolysis in the later stages of HMGECs differentiation should be elucidated in further studies. Furthermore, since the accumulation of lipid droplets is the sum of lipogenesis and lipolysis reactions, it will be necessary in future studies to investigate whether activated PKA phosphorylates lipolysis-related molecules, such as perilipin, ATGL, and HSL in HMGECs.

One of the most representative signals activating the PKA pathway is an increase in cAMP levels via beta-adrenergic receptors [22]. As various glaucoma therapeutics act by blocking beta-adrenergic receptors, it is clinically important to evaluate the effect of beta-receptor signaling on HMGEC differentiation. We evaluated whether ADRB2 affects subsequent PKA pathways as forskolin did. As shown in Figure 3 and Figure 4, the ADRB2 agonist salbutamol promoted the PKA pathway and subsequent PPARγ expression and lipid synthesis, whereas the ADRB2 antagonist timolol showed the opposite effect. Although it is now widely accepted that topical anti-glaucoma drugs aggravate MGD [19,20], the underlying mechanisms remain to be elucidated. Topical beta-blockers cause decreased tear production, tear film instability, corneal epithelial damage, and a reduction of conjunctival goblet cells [36,37,38,39,40,41]. Although these ocular surface changes may be the causes of MGD, evidence on whether a beta-blocker directly induces MGD is still lacking. Our findings might help understand why timolol worsens MGD in patients with glaucoma and the overall pathophysiology of MGD. Previous studies investigating the effect of timolol on HMGECs did not show a significant effect on lipid synthesis [23]. This discrepancy might be due to the different compositions of the differentiation media. Zhang et al. used azithromycin to enhance lipid synthesis in HMGECs; however, whether these differences have other effects on lipid synthesis signals requires further study. In addition, it is known that, not only the decrease in lipid production, but also the change in the lipid composition are important for the pathophysiology of MGD [42,43]. Therefore, future studies should investigate whether the lipid composition of HMGECs is also changed by the beta-blocker.

In conclusion, we demonstrated that the activation of the PKA pathway enhanced PPARγ expression and lipid synthesis in HMGECs. The physiological relevance of ADRB2 to the PKA pathway was also evaluated. The ADRB2 agonist salbutamol enhanced the PKA pathway and subsequent PPARγ expression and lipid synthesis, whereas the ADRB2 antagonist timolol had the opposite effect. These results show that the ADRB2-PKA pathway could be implicated in the pathogenesis of MGD and provide an understanding of how drugs, such as timolol, could aggravate MGD in patients. In addition, the ADRB2-PKA pathway may be a potential target for MGD treatment.

## 4. Materials and Methods

### 4.1. Cell Culture and Differentiation

The immortalized HMGEC—kindly provided by Dr. Sullivan (Schepens Eye Research Institute)—were cultured in keratinocyte serum-free medium (KSFM; Gibco, Grand Island, NY, USA) as previously described [13]. HMGECs were differentiated through culturing in DMEM-F12 (Lonza, Basel, Switzerland), supplemented with 10% (*v*/*v*) fetal bovine serum and 10 ng/mL epidermal growth factor (R&D Systems, Minneapolis, MN, USA), with or without drug compounds, according to each experiment. To assess the phosphorylation of CREB, chemicals were treated with fresh differentiation media after 2 h of incubation. To evaluate the PPARγ pathway and lipid accumulation, differentiation media were added on days 0 and 2. Forskolin (Sigma, St. Louis, MO, USA) was added to the differentiation medium at a final concentration of 1 µM. Salbutamol (Sigma) and timolol maleate (Sigma) were supplemented at final concentrations of 1 µM and 0.0002%, respectively.

### 4.2. Quantitative PCR Analysis

qPCR was performed as previously described [13]. Total RNA was isolated using Tri-RNA reagent (FAVORGEN, Ping-Tung, Taiwan), and 1 μg of total RNA was used to synthesize cDNA using RNA-to-cDNA EcoDryTM premix (TaKaRa, Shiga, Japan) according to the manufacturer’s protocol. Relative mRNA levels were evaluated in ViiA7 (Applied Biosystems, Foster City, CA, USA) using the SYBR Green PCR Master Mix (Applied Biosystems). Target gene expression was normalized to that of the housekeeping gene, glyceraldehyde 3-phosphate dehydrogenase (GAPDH). Primer sequences used in this study are listed in Table 1.

### 4.3. Immunoblot Analysis

Immunoblotting was performed as previously described [44]. The cells were grown in a 12-well plate, and the experiments were repeated at least three times. Cell lysates were prepared using a cell lysis buffer (Cell Signaling Technology, Danvers, MA, USA) containing 1 μM PMSF (Roche, Basel, Switzerland). The protein levels of the obtained lysates were quantified using BCA Protein Assay reagent (Thermo Fisher Scientific, Waltham, MA, USA). The cell lysates were diluted in a 5× sodium dodecyl sulfate (SDS) sample buffer and then separated on 10% SDS-PAGE gel. The separated proteins were transferred onto polyvinylidene fluoride membranes (Invitrogen, Carlsbad, CA, USA), which were then incubated with appropriate primary and secondary antibodies. Specific proteins were detected using rabbit anti-PPARγ (Cell Signaling Technology), mouse anti-SREBP-1 (Santa Cruz Biotechnology, Dallas, TX, USA), rabbit anti-CREB (Cell Signaling Technology), rabbit anti-phospho-CREB (ser133, Cell Signaling Technology), chicken anti-ADRB2 (Abcam, Cambridge, MA, USA), and mouse anti-GAPDH (Santa Cruz Biotechnology). Protein bands were detected using horseradish peroxidase-conjugated anti-chicken IgY (Abcam), anti-rabbit IgG (AbFrontier, Seoul, Korea), and anti-mouse IgG (AbFrontier) antibodies, and enhanced chemiluminescence (GE Healthcare, Little Chalfont, UK).

### 4.4. Oil Red O Staining and AdipoRed Assay

Oil Red O staining and AdipoRed assays were performed as previously described, with some modifications [13]. The cells were fixed with 4% paraformaldehyde (Biosesang, Seongnam, Korea) for 20 min at room temperature (RT) and washed three times with distilled water. Subsequently, 100% propylene glycol (Junsei Chemical, Tokyo, Japan) was added and left to stand for 2 min at RT. After complete removal of propylene glycol, the cells were stained with 0.3% Oil Red O (Sigma) solution for 1 h at room temperature. After the removal of the Oil Red O solution, 85% propylene glycol was applied for 2 min and washed twice with distilled water. To quantify the lipid droplets, the stained cells were eluted with 100% isopropanol (Millipore, Billerica, MA, USA), and the absorbance of the extracts was measured at 492 nm.

For the AdipoRed assay, the cells were grown in a 48-well plate in triplicate for each cycle. The cells were stained with AdipoRed (Lonza) according to the manufacturer’s instructions. After 10 min, the stained cells were placed in a fluorimeter (Varioskan Flash, Thermo Fisher Scientific) and the fluorescence was measured at excitation and emission wavelengths of 485 and 572 nm, respectively.

### 4.5. Confocal Immunocytochemistry

The differentiated cells, grown in the Nunc™ Lab-Tek™ II Chamber Slide™ system (Thermo Fisher Scientific), were fixed with 4% paraformaldehyde (Biosesang) for 30 min at room temperature and then washed twice with phosphate-buffered saline. The cells were stained by incubation with LipidTOX (Invitrogen) for 30 min at room temperature and mounted with gel/mount containing DAPI (Molecular Probes, Eugene, OR, USA). Fluorescence images were obtained using a Zeiss LSM 700 confocal microscope (Carl Zeiss, Oberkochen, Germany).

### 4.6. Statistical Analysis

The results of multiple experiments are presented as the mean ± standard error of the mean, and *p* < 0.05 was considered significant. All experiments were repeated at least three times. The results of the cell culture experiments were compared using Student’s *t*-test using GraphPad Prism 8 (GraphPad Software, Inc., La Jolla, CA, USA).

## Figures and Tables

**Figure 1 ijms-23-09478-f001:**
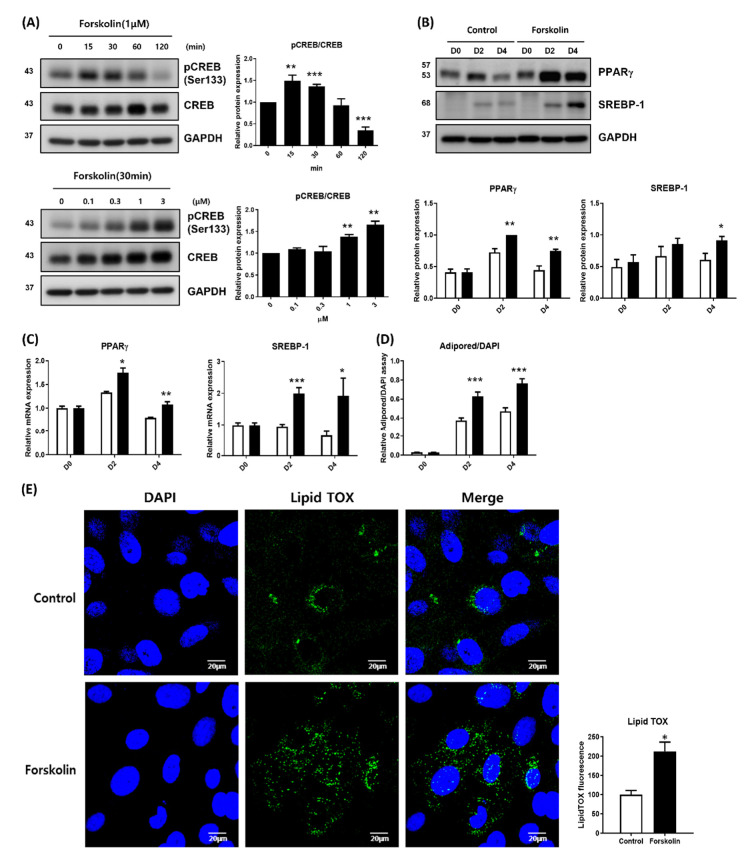
PKA activation enhanced lipid synthesis in human meibomian gland epithelial cells (HMGECs). (**A**) HMGECs were treated with forskolin, and the phosphorylation of CREB at ser133 residue was assessed by immunoblotting in serial and dose escalation manners. After treatment with forskolin (1 μM), the CREB phosphorylation was assessed with time. CREB phosphorylation was highest at 30 min after treatment and decreased thereafter. After forskolin was treated by concentration, CREB phosphorylation was observed 30 min later. CREB phosphorylation increased with forskolin concentration (the sum of 3 and 4 experiments, respectively). (**B**–**D**) HMGECs were cultured with or without forskolin (1 μM; forskolin was treated at day 0 and 2). (**B**,**C**) mRNA and protein expression of PPARγ and SREBP-1 were assessed by quantitative PCR and immunoblotting at day 0, 2, and 4 (the sum of 3 and 4 experiments respectively). (**D**,**E**) Lipid accumulation was measured by AdipoRed assay and LipidTOX assay at day 4, and the results of three experiments are summarized. Data are presented as the mean ± standard error of the mean. D, day after differentiation; * *p* < 0.05; ** *p* < 0.01; *** *p* < 0.001.

**Figure 2 ijms-23-09478-f002:**
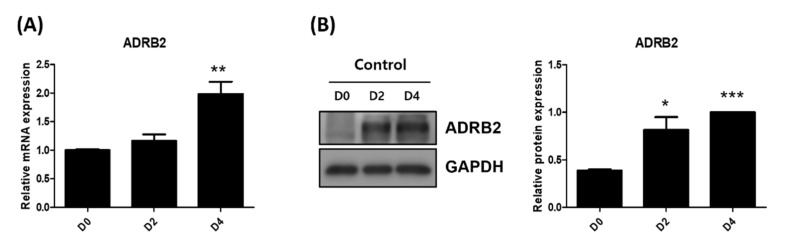
Differentiated HMGECs increased expression of beta 2-adrenergic receptor (ADRB2) (**A**) HMGECs were cultured with differentiation media from Day 0. mRNA (**A**) and protein expression (**B**) were measured by quantitative PCR and immunoblotting at day 0, 2, and 4, respectively. Data are presented as the mean ± standard error of the mean. D, day after differentiation; * *p* < 0.05; ** *p* < 0.01, *** *p* < 0.001.

**Figure 3 ijms-23-09478-f003:**
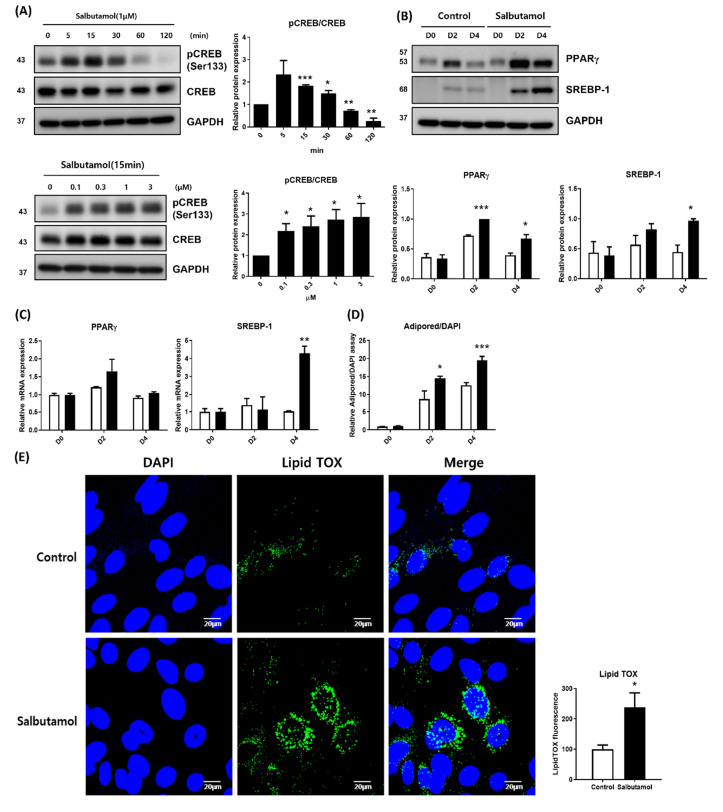
Beta 2-adrenergic agonist activates PKA signaling pathway, followed by promoted PPARγ expression and lipid droplet synthesis. (**A**) HMGECs were treated with salbutamol and phosphorylation of CREB at ser133 residue, which was assessed by immunoblotting in serial and dose escalation manners. After treatment with salbutamol (1 μM), the CREB phosphorylation was assessed with time. CREB phosphorylation was highest at 15 min after treatment and decreased thereafter. After salbutamol was treated by concentration, CREB phosphorylation was examined 15 min later. CREB phosphorylation increased with salbutamol concentration. (**B**–**E**) HMGECs were cultured with or without salbutamol (1 μM; salbutamol was treated at day 0 and 2). (**B**,**C**) Transcription and protein expression of PPARγ and SREBP-1 were assessed by quantitative PCR and immunoblotting at day 0, 2, and 4. (**D**,**E**) Lipid synthesis was measured by AdipoRed assay and LipidTOX assay at day 4. All results are the sum of three independent experiments. Data are presented as the mean ± standard error of the mean. D, day after differentiation; * *p* < 0.05; ** *p* < 0.01; *** *p* < 0.001.

**Figure 4 ijms-23-09478-f004:**
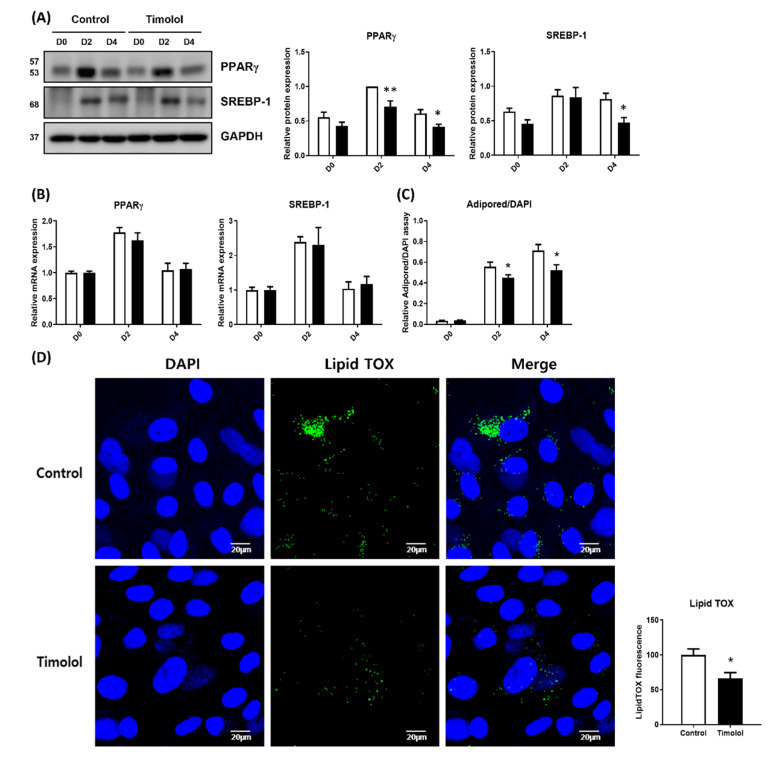
Beta 2-adrenergic antagonist inhibits lipogenesis in HMGECs. HMGECs were cultured with or without timolol (0.0002%; timolol was treated at days 0 and 2). (**A**,**B**) mRNA and protein expression levels of PPARγ and SREBP-1 were assessed by quantitative PCR and immunoblotting on days 0, 2, and 4 (the sum of 3 and 4 experiments, respectively). (**C**,**D**) Lipid synthesis was measured by AdipoRed assay and LipidTOX assay at day 4, and the results of three experiments are summarized. Data are presented as mean ± standard error of the mean. D, days after differentiation; * *p* < 0.05; ** *p* < 0.01.

**Figure 5 ijms-23-09478-f005:**
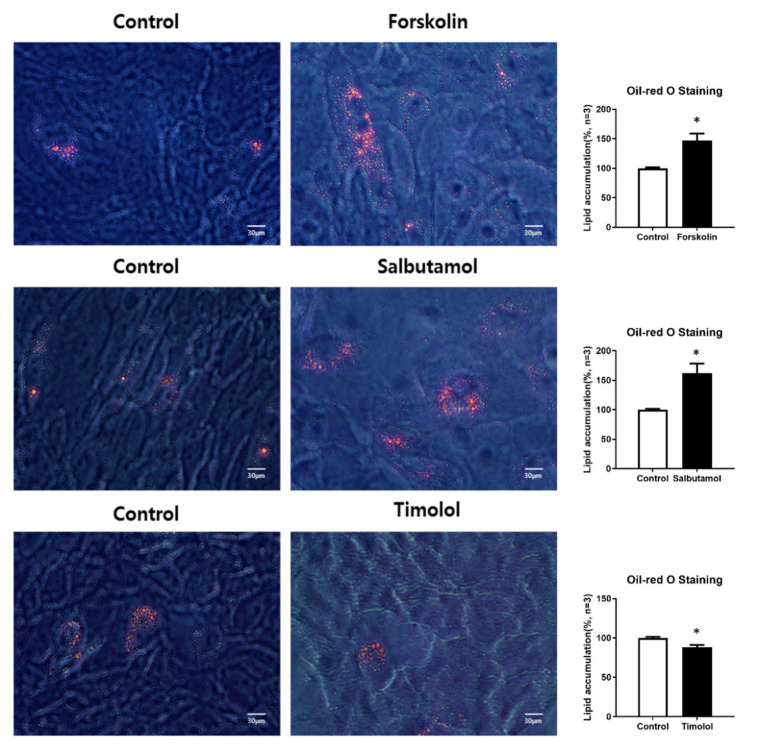
Lipid synthesis was examined by Oil Red O staining at day 4 after treatment with forskolin, salbutamol, and timolol. HMGECs were cultured with or without forskolin, salbutamol, and timolol, and lipid accumulation was assessed using Oil Red O staining on day 4. The results of three experiments are summarized in the right panels. Data are presented as the mean ± standard error of the mean; * *p* < 0.05.

**Table 1 ijms-23-09478-t001:** Primers used for the quantitative PCR analysis.

Gene	Forward (5′–3′)	Reverse (5′–3′)	Product Size (bp)
GAPDH	GAG TCA ACG GAT TTG GTC GT	GAC AAG CTT CCC GTT CTC AG	185
PPARγ	TTG CAG TGG GGA TGT CTC AT	TTT CCT GTC AAG ATC GCC CT	208
SREBP-1	GAG CTC AAG GAT CTG GTG GT	CCG ACA CCA GAT CCT TCA GA	175

## Data Availability

The data will be available from the corresponding author following reasonable request.

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
