# Peer review of "Activation of ADRB2/PKA Signaling Pathway Facilitates Lipid Synthesis in Meibocytes, and Beta-Blocker Glaucoma Drug Impedes PKA-Induced Lipid Synthesis by Inhibiting ADRB2"

_ijms, 2022, doi:10.3390/ijms23169478_

Round 1
Reviewer 1 Report
In this manuscript, the authors investigated the role of the beta2-adrenergic receptor (ADRB2) and protein kinase A (PKA) pathway in lipogenesis in human meibomian gland epithelial cells (HMGECs) and propose blockage of this pathway as a mechanism of meibomian gland dysfunction seen with the use of anti-glaucoma drugs targeting ADRB2.
The presented data through well-planned experiments demonstrates the involvement of ADRB2/PKA pathway in lipid synthesis in HMGECs.
Main comment:
As stated in the Discussion, the PKA pathway is also known to be involved in lipolysis. On page 8, lines 235-237, the authors state, “In our study, activation of the PKA pathway enhanced PPARγ expression and lipid synthesis rather than lipolysis in HMGECs.” However, lipolysis was not investigated in this study. Did the authors consider performing experiments to study the role of PKA pathway in lipolysis in HMGECs? Could they present that data to support their statement?
Other comments:
The title of the manuscript sounds contradictory. Can it be revised to inform about the study more accurately?
The heading of sections 2.4, “Cact-3 potently activates endogenous CFTR chloride channels and increases tear volume in mice” is not relevant to the present study and must be changed.
The Discussion is a bit difficult to follow at times, particularly paragraphs 2 and 3, and is repetitive. Can the Discussion be improved, and repetition avoided, particularly in the last two paragraphs?
In Figures 1A and 3A legends, please clarify that the effect of forskolin and salbutamol, respectively, at different time-points and of different doses was assessed.
In Figures 1, 3, 4 and 5, please indicate the results of how many different experiments have been summarized in each case.
Abstract, lines 26 and 27, “(which is at the upper level of the PKA signaling system)”, probably could be “(which is upstream of PKA signaling)”.
Line 332, please confirm if cells were washed with 4% formaldehyde after fixation with paraformaldehyde.
Author Response
Point-by-point response
Reviewer #1:
In this manuscript, the authors investigated the role of the beta2-adrenergic receptor (ADRB2) and protein kinase A (PKA) pathway in lipogenesis in human meibomian gland epithelial cells (HMGECs) and propose blockage of this pathway as a mechanism of meibomian gland dysfunction seen with the use of anti-glaucoma drugs targeting ADRB2.
The presented data through well-planned experiments demonstrates the involvement of ADRB2/PKA pathway in lipid synthesis in HMGECs.
[Response] We thank the reviewer for this careful review of the manuscript and for these kind words.
Main comment:
- As stated in the Discussion, the PKA pathway is also known to be involved in lipolysis. On page 8, lines 235-237, the authors state, “In our study, activation of the PKA pathway enhanced PPARγ expression and lipid synthesis rather than lipolysis in HMGECs.” However, lipolysis was not investigated in this study. Did the authors consider performing experiments to study the role of PKA pathway in lipolysis in HMGECs? Could they present that data to support their statement?
[Response] Thank you for the critical comments. Lipid synthesis is the process of converting glycerol and free fatty acids into triacylglycerols, resulting in the accumulation of lipid droplets in the cytoplasm. Lipolysis is the opposite processes and results in a decrease in lipid droplets in the cytoplasm. According to our experimental results, since the lipid droplets of HMGECs accumulated with activation of the PKA pathway, we simply thought and described that the PKA pathway acted more on lipid synthesis rather than lipolysis in our manuscript.
However, since the accumulation of lipid droplets is the sum of lipogenesis and lipolysis reactions, it would have been a more accurate study if we had investigated whether the activated PKA phosphorylates lipolysis-related molecules such as perilipin, adipose triglyceride lipase (ATGL) and hormone-sensitive lipases (HSL).
Unfortunately, we didn't think about it that far during our experiments. Therefore, there is no experimental data related to this concern. In future research, we will remember the opinions of the reviewers and plan the research in detail. In addition, we describe these matters in the revised manuscript as below.
“Furthermore, since the accumulation of lipid droplets is the sum of lipogenesis and lipolysis reactions, it will be necessary in future studies to investigate whether activated PKA phosphorylates lipolysis-related molecules such as perilipin, ATGL and HSL in HMGECs.”
Other comments:
- The title of the manuscript sounds contradictory. Can it be revised to inform about the study more accurately?
[Response] Thank you for your comments. As suggested by reviewer, we changed the title as follows:
“Activation of ADRB2/PKA Signaling Pathway Facilitates Lipid Synthesis in Meibocytes, and Beta-Blocker Glaucoma Drug Impedes PKA-induced Lipid Synthesis by Inhibiting ADRB2.”
- The heading of sections 2.4, “Cact-3 potently activates endogenous CFTR chloride channels and increases tear volume in mice” is not relevant to the present study and must be changed.
[Response] We are sorry to have made such a mistake. The title of the chapter has been changed as follows:
“2.4 Beta 2-adrenergic antagonist inhibits lipogenesis in HMGECs”
- The Discussion is a bit difficult to follow at times, particularly paragraphs 2 and 3, and is repetitive. Can the Discussion be improved, and repetition avoided, particularly in the last two paragraphs?
[Response] In response to the reviewer’s comment, we removed some repetitions and revised the paragraphs.
“PKA is a tetramer enzyme that includes two regulatory subunits and two catalytic subunits. When cAMP binds….”
“For CREB, a well-characterized PKA substrate, when the CREB Ser133 residue is phosphorylated, CREB binds to the cAMP response element and initiates target gene transcription. and phosphyorylation of the CREB Ser133 residue is pivotal for the transcriptional activity of CREB. Activated CREB can bind to the cAMP response element in the promoter region of target genes and initiate genes transcription”
“After confirming that differentiated HMGECs express ADRB2, We evaluated whether ADRB2 signaling affects subsequent PKA pathways as forskolin did.”
“In Conclusion, we demonstrated that the activation of the PKA pathway enhanced PPARγ expression and lipid synthesis in HMGECs. The physiological relevance of ADRB2 to the PKA pathway was also evaluated. by treatment with ADRB2 agonists and antagonists. The ADRB2 agonist salbutamol enhanced the PKA pathway and subsequent PPARγ expression and lipid synthesis, whereas the ADRB2 antagonist timolol had the opposite effect. These results show that the ADRB2-PKA pathway could be implicated in the pathogenesis of MGD and provide an understanding of how drugs such as timolol, which affect the ADRB2-PKA pathway, could aggravate MGD in patients with glaucoma. In addition, the ADRB2-PKA pathway may be a potential target for MGD treatment.”
- In Figures 1A and 3A legends, please clarify that the effect of forskolin and salbutamol, respectively, at different time-points and of different doses was assessed.
[Response] Thank you for the helpful comments. We described the effect of forskolin and salbutamol in detail as follows:
“Figure 1A Legend
After treatment with forskolin(1μM), the CREB phosphorylation was assessed with time. CREB phosphorylation was highest at 30 min after treatment and decreased thereafter. After forskolin was treated by concentration, CREB phosphorylation was observed 30 minutes later. CREB phosphorylation increased with forskolin concentration.”
“Figure 3A Legend
After treatment with salbutamol(1μM), the CREB phosphorylation was assessed with time. CREB phosphorylation was highest at 15 min after treatment and decreased thereafter. After salbutamol was treated by concentration, CREB phosphorylation was observed 15 minutes later. CREB phosphorylation increased with salbutamol concentration.”
- In Figures 1, 3, 4 and 5, please indicate the results of how many different experiments have been summarized in each case.
[Response] In response to the reviewer’s comment, we added the total number of experiments in each figure legend as follows:
“Figure 1 legend
… After forskolin was treated by concentration, CREB phosphorylation was observed 30 minutes later. CREB phosphorylation increased with forskolin concentration (the sum of 3 and 4 experiments, respectively). (B-D) HMGECs were cultured with or without forskolin (1μM; Forskolin was treated at day 0 and 2). (B-C) mRNA and protein expression of PPARγ and SREBP-1 were assessed by quantitative PCR and immunoblotting at day 0, 2 and 4 (the sum of 3 and 4 experiments respectively). (D-E) Lipid accumulation was measured by Adipored assay and LipidTOX assay at day 4 and the results of three experiments are summarized. Data are presented as the mean ± standard error of the mean. D, day after differentiation; *P < 0.05; **P < 0.01; ***P < 0.001.
Figure 3 legend
…All results are the sum of three independent experiments. Data are presented as the mean ± standard error of the mean. D, day after differentiation; *P < 0.05; **P < 0.01; ***P < 0.001.
Figure 4 legend
… (A-B) mRNA and protein expression levels of PPARγ and SREBP-1 were assessed by quantitative PCR and immunoblotting on days 0, 2, and 4 (the sum of 3 and 4 experiments, respectively). (C-E) Lipid synthesis was measured by AdipoRed assay and LipidTOX assay at day 4, and the results of three experiments are summarized. Data are presented as mean ± standard error of the mean. D, days after differentiation; *P < 0.05; **P < 0.01.
Figure 5 legend
… The results of three experiments are summarized in the right panels. Data are presented as the mean ± standard error of the mean; *P < 0.05.”
- Abstract, lines 26 and 27, “(which is at the upper level of the PKA signaling system)”, probably could be “(which is upstream of PKA signaling)”.
[Response] Thank you for the comment. We have made changes to the manuscript as you pointed out.
- Line 332, please confirm if cells were washed with 4% formaldehyde after fixation with paraformaldehyde.
[Response] Thank you for your detailed comments. The cells were washed with Dulbecco's phosphate buffered saline after fixation with paraformaldehyde. We have rewritten the revised manuscript as follows:
“The differentiated cells, grown in Nunc™ Lab-Tek™ II Chamber Slide™ system (Thermo Fisher Scientific), were fixed with 4% paraformaldehyde (Biosesang) for 30 min at room temperature and then washed twice with phosphate buffered
Reviewer 2 Report
Jun and colleagues evaluated the role of the beta2-adrenergic receptor (ADRB2)/protein kinase A (PKA) pathway in lipid synthesis in cultured human meibomian gland epithelial cells. The topic is very interesting and the manuscript has been written very well. The study design and methods are appropriate and described clearly. The results are presented very well and support the conclusions. I have a few suggestions and comments for the authors.
General comments:
1. Lipid synthesis is not the only mechanism involved in MGD. There are types of hypersecretory MGDs that lipid composition is altered. This study only evaluates the amount of lipid synthesis, but not its composition. Hence, the clinical translation of these results should be assessed in future studies.
2. The title highlights the beta-blocker glaucoma drug-induced MGD. However, it is not mentioned in the introduction and is only briefly discussed in the discussion. Only 2 papers have been cited, both show that prostaglandin analogues equally cause MGD. Please explain the role of beta-blocker (both topical and systemic) in MGD in the introduction. Also, the discussion on beta-blocker-induced MGD should be expanded.
3. The authors have used long, complex sentences on several occasions (e.g., Lines 26-29) that can cause confusion. Please consider revising the sentences (e.g., splitting them into two or more sentences) to make it more clear to the readers.
4. I suggest that the authors appreciate the role of lipid composition in MGD and the potential role of beta-blockers in altering the meibum composition. This can be added as a potential limitation of this study that should be explored in future research.
Minor comments:
1. Line 132: "Subsequently, cells were treated with and salbutamol on days..." There is something missing before "and". Please correct.
2. Line 150: "Cact-3 potently activates endogenous CFTR chloride channels and increases tear volume in mice". I suppose this belongs to another paper? Please correct.
3. Line 208: "The role of the PKA pathway in meibomian gland dysfunction and dry eye syndrome is limited." Should this be "Evidence on the role of the PKA pathway ... is limited"? Please consider revising.
Author Response
Point-by-point response
Reviewer #2:
Jun and colleagues evaluated the role of the beta2-adrenergic receptor (ADRB2)/protein kinase A (PKA) pathway in lipid synthesis in cultured human meibomian gland epithelial cells. The topic is very interesting and the manuscript has been written very well. The study design and methods are appropriate and described clearly. The results are presented very well and support the conclusions. I have a few suggestions and comments for the authors.
[Response] We would like to thank the reviewers for their thoughtful comments and efforts toward improving out manuscript. In the following, we highlight comments specific to each reviewer and our effort to address these concerns.
General comments:
- Lipid synthesis is not the only mechanism involved in MGD. There are types of hypersecretory MGDs that lipid composition is altered. This study only evaluates the amount of lipid synthesis, but not its composition. Hence, the clinical translation of these results should be assessed in future studies.
[Response] Thank you for the critical comments. Lipid composition of meibum could affect the stabilization of the tear film lipid layer (Butovich et al. J Lipid Res 2009;50:2471–85; Jun et al. Am J Ophthalmol. 2022 Aug;240:37-50) In addition, lipid composition can vary depending on various circumstances such as the presence of MGD, age, and serum lipid levels (Suzuki et al, Ocul Surf. 2021 Oct 16;S1542-0124(21)00120-8; Yoo et al, J Clin Med. 2022 Jul 11;11(14):4010).
As you pointed out, lipid composition studies of ADRB2/PKA pathway in HMGECs would be valuable preclinical data before proceeding with experiments using human samples, although the lipid composition in HMGECs under serum treatment does not reflect lipid profile of human meibum in vivo (Hampel et al. PLoS One. 2015 Jun 4;10(6):e0128096). we describe these concerns in the revised manuscript as below.
“In addition, it is known that not only the decrease in lipid production but also the change in the lipid composition are important for the pathophysiology of MGD (Suzuki et al, Ocul Surf. 2021 Oct 16;S1542-0124(21)00120-8; Jun et al. Am J Ophthalmol. 2022 Aug;240:37-50) Therefore, future studies should investigate whether the lipid composition of HMGECs is also changed by the beta-blocker.”
- The title highlights the beta-blocker glaucoma drug-induced MGD. However, it is not mentioned in the introduction and is only briefly discussed in the discussion. Only 2 papers have been cited, both show that prostaglandin analogues equally cause MGD. Please explain the role of beta-blocker (both topical and systemic) in MGD in the introduction. Also, the discussion on beta-blocker-induced MGD should be expanded.
[Response] Thank you for the important comments. It is well known that topical antiglaucoma drugs could contribute to induce MGD. However, most studies have not been analyzed according to the type of glaucoma drug and studies on pathogenesis are more limited. Since the analysis of beta blocker was relatively good in those two papers referenced in discussion, those papers were chosen and added to the body of the article. In particular, the relationship between systemic beta-blockers and MGD still remained to be elucidated.
We added these concerns to the introduction and discussion.
“Introduction
In addition, it is clinically important to study the PKA pathway because eye drops that modulate the PKA pathway are widely used in glaucoma patients. Beta blocker is one of the widely used anti-glaucoma eye drops and is known to cause damage to the ocular surface including MGD. (Arita et al, Graefes Arch Clin Exp Ophthalmol. 2012 Aug;250(8):1181-5; Uzunosmanoglu et al. Cornea. 2016 Aug;35(8):1112-6; Zhou et al. Ophthalmol Ther. 2022 Aug 9) However, the pathophysiology of MGD-inducing effect is remained to be elucidated.”
“Discussion
Topical beta-blockers cause decreased tear production, tear film instability, corneal epithelial damage, and reduction of conjunctival goblet cells. (Ishibashi et al. Cornea. 2003 Nov;22(8):709-15; Rolle et al. BMC Ophthalmol. 2017 Aug 3;17(1):136; Trope et al. J Ocul Pharmacol. 1988;4(4):359–66; Nam et al. J Clin Med. 2021 Aug 5;10(16):3464; Yuan et al. BMC Ophthalmol. 2021;21(1):419; Ciancaglini et al. Eur J Ophthalmol. 2008;18(3):400–7.) Although these ocular surface changes may be the causes of MGD, evidence on whether beta-blocker directly induces MGD are still lacking.”
- The authors have used long, complex sentences on several occasions (e.g., Lines 26-29) that can cause confusion. Please consider revising the sentences (e.g., splitting them into two or more sentences) to make it more clear to the readers.
[Response] Thank you for the helpful comments. We revised the sentences as follows:
“When treated with agonists of ADBR2 (upstream of the PKA signaling system), PPARγ expression and lipid synthesis were enhanced in HMGECs. The ADRB2 antagonist timolol showed the opposite effect.”
- I suggest that the authors appreciate the role of lipid composition in MGD and the potential role of beta-blockers in altering the meibum composition. This can be added as a potential limitation of this study that should be explored in future research
[Response] We appreciated your careful review. Please see our answer to your comment #1, we describe these concerns in the revised manuscript as below.
“In addition, it is known that not only the decrease in lipid production but also the change in the lipid composition are important for the pathophysiology of MGD (Suzuki et al, Ocul Surf. 2021 Oct 16;S1542-0124(21)00120-8; Jun et al. Am J Ophthalmol. 2022 Aug;240:37-50) Therefore, future studies should investigate whether the lipid composition of HMGECs is also changed by the beta-blocker.”
Minor comments:
- Line 132: "Subsequently, cells were treatedwith and salbutamolon days..." There is something missing before "and". Please correct.
[Response] Thank you for the comment. we removed “and” in the revised manuscript.
- Line 150: "Cact-3 potently activates endogenous CFTR chloride channels and increases tear volume in mice". I suppose this belongs to another paper? Please correct.
[Response] We are sorry to have made such a mistake. The title of the chapter has been changed as follows:
“2.4 Beta 2-adrenergic antagonist inhibits lipogenesis in HMGECs”
- Line 208: "The role of the PKA pathway in meibomian gland dysfunction and dry eye syndrome is limited." Should this be "Evidence on the role of the PKA pathway ... is limited"? Please consider revising.
[Response] Based on the comments of the reviewers, the sentence has been modified as follows.
“The role of the PKA pathway in meibomian gland dysfunction and dry eye disease is not well understood.”